# Understanding travel intention formation in government culture and tourism TikTok accounts: An integration of the SOR model and emotion appraisal theory

Yuan Sun, Bin Wen *

School of History, Culture and Tourism, Huaibei Normal University, Huaibei, Anhui, China

* yellowriverwenbin@163.com

## Abstract

This study draws on the Stimulus–Organism–Response (SOR) model and emotion appraisal theory to examine how information quality and service quality of government culture and tourism TikTok accounts are linked to audience travel intentions. It identifies mediating mechanisms involving destination image perception and positive emotions. Using survey data from 406 respondents, a structural equation model is constructed for analysis. The results indicate that information quality is positively associated with audience travel intentions and is also linked to it indirectly through destination image perception and positive emotions. Service quality is not directly associated with audience travel intentions but is indirectly associated with them through destination image perception and positive emotions. In addition, both information quality and service quality are related to audience travel intentions through the serial mediation of destination image perception and positive emotions. Based on these findings, the study suggests an operational approach that emphasizes enhancing information quality, integrating intelligent service-interaction functions, and adopting an operational framework that moves from cognitive construction to emotional reinforcement. These suggestions provide practical insights for social media marketing in government tourism and culture departments.

## 1. Introduction

In recent years, the rapid expansion of short video platforms has significantly influenced destination marketing systems. Douyin, the Chinese mainland version of TikTok, is one of the largest short video platforms in China. For consistency with the international literature, Douyin is hereafter referred to as TikTok in this study. Given its immersive interface, interactive features, and algorithm-driven recommendation system, TikTok is widely regarded as an important digital platform for destination image dissemination. In response to this development, government culture and tourism

**Data availability statement:** All relevant data are within the manuscript and its Supporting information files.

**Funding:** This study is funded by the "Anhui Provincial Philosophy and Social Science Planning Project". Project Approval Number: AHSKYQ2023D015.

**Competing interests:** The authors have declared that no competing interests exist.

departments have actively established official TikTok accounts. For instance, the Henan culture and tourism TikTok account gained over a million followers in just four days, and the Xi'an culture and tourism TikTok account is reported to have enhanced the city's image by leveraging the cultural IP "The Secret Box of the Prosperous Tang Dynasty" to engage younger audiences. These government culture and tourism Tik-Tok accounts (GCTTAs) have been increasingly positioned as key channels in destination branding. Unlike commercial brands or ordinary content creators, GCTTAs are not merely content producers. They perform a composite role encompassing information dissemination and public communication [1], commonly engaging in interactive functions such as responding to comments, supplementing activity information, and addressing complaints and suggestions. Accordingly, audience evaluations of their communication effectiveness are shaped not only by the attractiveness of the content itself, but also by their perceptions of the accounts' public service performance. However, existing research on short video tourism marketing has paid limited attention to these dual mechanisms.

Current research on short video marketing has largely evolved around three major paradigms. The first is the experience-oriented paradigm, grounded in flow theory, which posits that the audiovisual stimulation of short videos creates immersive presence, inducing flow experiences that enhance audience travel intentions [2,3]. The second is the cognitive-behavioral paradigm, based on the SOR model, which focuses on how video features stimulate users' perceptions and emotions to influence destination decision-making [4,5]. The third is the technology-information processing paradigm, which employs the Technology Acceptance Model (TAM), Information Adoption Model (IAM), Elaboration Likelihood Model (ELM), and their extensions to examine how users process technological attributes and informational features of social media platforms, leading to shifts in attitudes and behaviors [6–8]. Across these three paradigms, stimuli are primarily conceptualized as the technological attributes of short video platforms or the characteristics of information content and its modes of presentation. These stimuli include the narrative transportation effect of short videos [9], application-related features [10], and dimensions of information quality such as comprehensiveness, relevance, timeliness, and source expertise [11]. Additionally, some studies have focused specifically on the narrative structure of short videos and the characters in short-form videos [12], the visual and linguistic framing employed in tourism promotional short videos [13]. Although some research has considered the quality of tourism information on social media, it often does not specify the source of the content under examination [14,15]. This lack of specificity regarding content source may limit the clarity and robustness of the conclusions. In addition, some studies have examined issues related to the reliability, efficiency, responsiveness, and perceived trust of government online services within the e-government literature [16–18]. However, e-government primarily functions to handle specific affairs, and this differs fundamentally from the functions and communication logic of government social media in tourism marketing. Therefore, the unique mechanisms underlying the relationships among information quality, service quality, and user behavioral responses in the context of government tourism communication remain insufficiently explored and warrant further investigation.

To address the identified research gap, this study draws on the SOR model and examines both information and service stimuli at the stimulus level to explain the distinctive mechanisms predicting audience travel intentions of GCTTAs. The SOR model originates from the Stimulus–Response paradigm in behaviorism, which conceptualizes behavior as an observable and measurable reaction to external environmental stimuli [19]. However, this paradigm does not account for the internal states of the organism that mediate responses to stimuli. To address this limitation, the SOR model incorporates internal psychological processes into the explanation of behavioral outcomes. It posits that environmental cues perceived by individuals activate internal evaluation and psychological processing states, which subsequently lead to positive or negative behavioral responses [20]. The organism construct originally referred primarily to emotional states, but it has since been extended to encompass both cognitive and emotional dimensions of psychological processing [21]. The SOR framework has become a useful lens for understanding the relationships among external inputs, emotional states, and behavioral outcomes [22]. However, the SOR model does not fully explicate the psychological processes through which emotions may emerge from cognitive processing. The emotion appraisal theory provides a critical complement for uncovering this internal mechanism. Arnold argued that emotions arise from individuals' subjective cognitive appraisal of the meaning of external stimuli [23]. Lazarus further argued that emotions arise from an individual's appraisal of the relevance of a stimulus to personal well-being within a specific context, the appraisal process is shaped by both situational factors and the individual's psychological background [24]. Accordingly, emotions are not direct reactions to stimuli, but psychological outcomes generated after cognitive appraisal. Regarding how emotions influence behavior, Arnold proposed the action sequence model, suggesting that emotions activate motivational tendencies that subsequently guide behavioral choices [25]. Subsequent research has conceptualized this mechanism as a sequential process in which cognitive appraisal elicits emotional responses, which in turn lead to behavioral outcomes [26]. Moors noted that the emotion appraisal theory has become one of the central theoretical frameworks for explaining the relationship between cognitive processing and emotional generation, with its key contribution lying in clarifying the mediating role of emotions between stimuli and behavior [27].

Based on the above theoretical integration, this study conceptualizes the organism construct in the context of GCTTAs as a psychological processing system composed of both cognitive appraisal and emotional response. Within this system, information quality and service quality, as external stimuli, are not only directly associated with audience travel intentions but also indirectly associated with them through destination image perception and positive emotions. In addition, higher levels of destination image perception are associated with stronger positive emotions, indicating a possible sequential mediation structure. Accordingly, a multi-path relational structure is proposed within the SOR model, in which parallel and sequential effects coexist, as illustrated in Fig 1.

This study makes three theoretical contributions to the field of government social media tourism marketing. First, this study extends existing research on service quality by demonstrating that its behavioral effects are contingent upon communication context. Prior studies, largely grounded in commercial marketing settings, tend to conceptualize information quality and service quality as parallel drivers that directly influence consumers' behavioral intentions. However, the present findings challenge this assumption by showing that, in the context of GCTTAs, the two stimuli operate through structurally different pathways. Specifically, information quality exerts both direct and indirect associations with travel intention, whereas service quality operates entirely through mediators. This fully mediated structure suggests that service quality in government culture and tourism departments' social media marketing does not function as an immediate behavioral driver, but rather as a relational signal that requires cognitive and emotional interpretation. This distinction advances existing theoretical perspectives by demonstrating that service quality is not a universal direct predictor of audience travel intention, but a context-contingent factor whose effects are jointly shaped by the nature of the communication source, the objectives of communication, and the characteristics of the social media platform. This study provides a more nuanced understanding of the mechanisms through which service quality operates in the context of government culture and tourism departments' social media marketing. Second, this study introduces the Heuristic–Systematic Model (HSM) as a complementary theoretical perspective to explain how different types of stimuli are associated with audience travel intentions

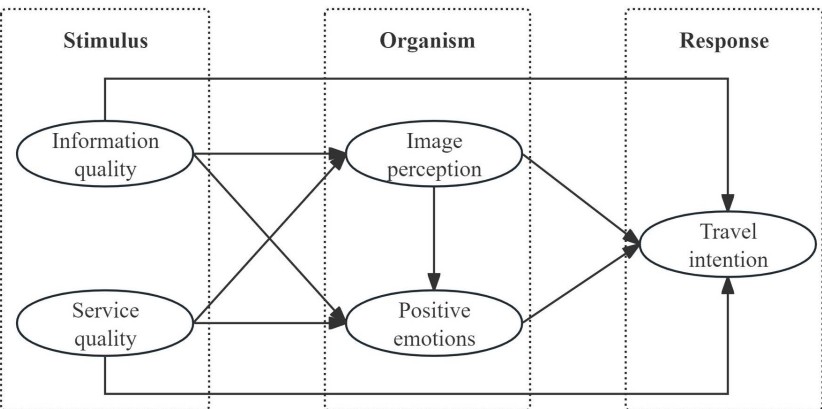

**Fig 1. Conceptual framework.**

in the government culture and tourism departments' social media marketing context. According to the HSM, information quality functions as a content-based cue that is directly associated with audience travel intentions via heuristic processing, and indirectly associated with travel intentions via systematic processing through cognitive and emotional evaluations.

In contrast, service quality is more likely to be perceived as a signal of the expertise, authority, and responsiveness of government culture and tourism departments. Its association with audience travel intentions primarily involves systematic processing and operates indirectly by strengthening destination image perception and positive emotions. The finding provides a new perspective for distinguishing between transaction-oriented service quality and public relationship-oriented service quality. Third, this study advances the structural development of the SOR framework in the public communication context. Existing SOR research often treats the organism as a single emotional or perceptual state and pays limited attention to its internal structure. By introducing emotion appraisal theory, this study divides the organism into two components. The first is cognitive evaluation, represented by destination image perception. The second is emotional experience, represented by positive emotions. The results provide empirical evidence for a sequential association, indicating that destination image perception is positively associated with positive emotions and follows a cognition-to-emotion progression. This mechanism deepens the explanation of internal psychological processes in the SOR framework and extends the model from a stimulus emotion response structure to a stimulus cognition emotion response structure. This refinement deepens the explanation of internal psychological processes within the SOR model and enhances its applicability in government culture and tourism departments' social media marketing research.

## 2. Literature review and research hypotheses

### 2.1. The impact of information quality on audience travel intentions

Tourism is essentially a phenomenon driven by the continuous flow and interpretation of information [28]. Travel decisions are usually made before the actual destination experience, and consumers are generally unable to fully verify product quality prior to actual consumption. Accordingly, information can be viewed as a key link between potential tourists and destinations. Parvaneh et al. [29] pointed out that effective information search can reduce risk and uncertainty in travel decision making and enhance decision confidence. However, the extent to which information reduces perceived risk largely depends on its quality. High-quality information helps users better understand products or services, access relevant support, and make more informed decisions [30]. Information quality refers to the audience's subjective evaluation of whether the characteristics of information meet their needs and intended use [31]. It is commonly conceptualized as a multidimensional construct, encompassing dimensions such as accuracy, consistency, interpretability, timeliness, and

completeness [32,33]. As research has developed, the dimensions of information quality have been updated according to specific research contexts. For example, Filieri [34] measures the information quality of electronic word of mouth using depth, factuality, breadth, relevance, and credibility. Considering the context of GCTTAs, this study measures short video information quality from three dimensions: content quality, utility quality, and presentation quality. Content quality refers to the extent to which the transmitted content is consistent with objective facts [31], which helps maintain the authority of government departments. Utility quality refers to the practical value of short video content for users' tourism consumption decisions, aiming to reduce their information search cost. Presentation quality refers to whether the short video content is concise, fluent, and easy to understand, emphasizing the transformation of official narratives into immersive experiences through audiovisual language.

Numerous studies suggest that influential online travel information on social media platforms is associated with users' travel purchase intentions, visit intentions, and word-of-mouth recommendations, thereby increasing the likelihood of purchasing or visiting [4,35,36]. However, most studies examine the role of information quality generated by tourism enterprises, such as OTA platforms and hotels, key opinion leaders, and ordinary users in relation to tourist decision-making [37–39], while studies on GCTTAs are limited. Unlike business-generated content or user-generated content, government short videos can be viewed as a key channel through which government departments disseminate authoritative information. Accordingly, the core function of GCTTAs is to provide the public with authoritative and comprehensive culture and tourism information, meeting the information needs of potential tourists before travel decisions. Guerrero-Rodríguez et al. [40] noted that official destination promotional videos present the destination positively to attract tourists, which can increase users' visit intentions. Based on this, the study proposes the following hypothesis:

**H1:** The information quality of tourism short videos of GCTTAs is positively associated with audience travel intentions.

## 2.2. The impact of service quality on audience travel intentions

Service quality is the core concept in the field of service marketing. Parasuraman et al. [41] pointed out that service quality is the gap between customers' expectations and their actual experiences, which is the most widely accepted definition of service quality. They developed the SERVQUAL scale, a five-dimensional framework comprising reliability, responsiveness, assurance, tangibles, and empathy, measured through 22 paired Likert-type items. The SERVQUAL model established the theoretical foundation for measuring offline service quality and has been widely used to explain outcome variables such as customer satisfaction, loyalty, and behavioral intention. As service contexts have shifted to online environments, service quality research has developed into the electronic service quality paradigm. Unlike offline settings, which emphasize physical contact and personal service, online services rely more on system stability, efficient feedback, and the presentation of information through technological and content-related features. For example, Iliachenko [42] noted that in online travel purchasing contexts, the interactivity and information characteristics of tourism websites are expected to have a strong influence on users. Chan et al. [43] found that hotel website functions and credible reviews increase consumer engagement and booking intention. Several studies have also examined service quality in the field of e-government and proposed the e-GovQual model, which includes service usability, ease of use, security, trust, information quality, and public support [16,17,44]. However, existing research mainly focuses either on commercial platforms and transaction decisions or on public affairs platforms. Limited attention has been given to government culture and tourism departments' social media accounts that combine destination image promotion and public communication functions.

**To address this gap,** building on prior research and considering the characteristics of GCTTAs, this study conceptualizes service quality as a multidimensional construct comprising reliability, responsiveness, and assurance, and treats it as a key antecedent of audience travel intentions. Unlike information quality, which focuses on the evaluation of content attributes of short videos, service quality emphasizes audience perceptions of the overall performance of government tourism departments in delivering information and services via the TikTok platform. This perception represents an account-level evaluation. Specifically, responsiveness refers to the timeliness of replying to audience comments, the accessibility of

feedback on suggestions, and the availability of interactive service functions, reflecting the efficiency and supportiveness of GCTTAs in user interactions. Assurance captures audience perceptions of the GCTTAs' trustworthiness, professional competence, and service attitude, thereby reflecting the credibility of the government culture and tourism departments as a public service provider. Reliability is commonly defined as users' trust in the accuracy and timeliness of the service delivery process [17]. In this study, it specifically refers to audiences' overall perception and evaluation of the stability, timeliness, and trustworthiness demonstrated by government culture and tourism departments in delivering information and services via the TikTok platform. Based on this analysis, the following hypothesis is proposed:

**H2:** The service quality of GCTTAs is positively associated with audience travel intentions.

## 2.3. The mediating role of destination image perception

Tasci and Gartner [45] stated that destination image formation is the mental representation of a destination formed after individuals select and process information cues from various image formation agents. This definition highlights that information sources and their characteristics are closely associated with image construction. Early studies focused on the role of traditional media in shaping destination image and argued that media shape audience perceptions of destinations through visual symbols and narrative framing [46]. With the rise of social media, online tourism information quality has become an important factor in shaping the online tourism market and promoting destination image formation. Kim et al. [30] developed a multidimensional model of online tourism information quality in social media contexts and examined how content cues and non-content cues are positively related to users' impressions of destinations. A positive destination image plays a critical role in post-visit evaluation and future travel intentions [47,48]. This implies that destination image perception serves as a mediating mechanism linking information quality and travel intention, which represents an important mechanism in tourism communication. In terms of media format, prior research has suggested that promotional videos are associated with tourists' perceptions of destination image [49]. Compared with text and image-based information, short videos integrate multiple elements such as visuals, color, music, text, and special effects. Well-designed short videos influence users' perception of destination image and travel intention through a combined sensory effect [50].

In addition to information-related factors, service quality has been identified as an important determinant of destination image. Prior research has widely examined the relationship between service quality and destination image [51,52]. However, existing studies on service quality mainly focus on traditional tourism enterprises or offline service settings, emphasizing the role of assurance, responsiveness, tangible facilities, empathy, reliability, and quality of directions in shaping destination image during service encounters [53]. In contrast, GCTTAs do not involve face-to-face service interactions, and audiences evaluate service quality mainly through online information services and interaction experiences, such as the timeliness of comment responses and the effectiveness of issue resolution. Nevertheless, individuals still integrate perceived service cues during destination-related contact and construct destination image at the cognitive level [54]. Accordingly, in the context of GCTTAs, higher levels of interaction and service support are associated with more favorable cognitive evaluations of the destination image. Prior research has shown that destination image is able to partially mediate the effect of service quality on travel intention [55]. Based on this analysis, the following hypotheses are proposed:

**H3:** Destination image perception mediates the relationship between information quality of GCTTAs and audience travel intentions.

**H4:** Destination image perception mediates the relationship between service quality of GCTTAs and audience travel intentions.

## 2.4. The mediating role of positive emotions

In tourism research, positive emotions are often described as feelings of pleasure, affection, and surprise [56]. Within the SOR framework, positive emotions represent an internal organismic state that links environmental stimuli and behavioral responses. Positive emotions arise from multiple situational and cognitive factors. First, at the level of situational stimuli,

the physical environment and spatial experience created by offline tourism destinations are observed to be important stimulating factors for the audience's positive emotions [57]. Similarly, the media environment created by online tourism short videos is significantly associated with audience emotional responses [58]. Second, at the level of service stimuli, social service environments (including staff emotional expression and customer interaction atmosphere), perceived service quality, and interactions between tourists have been found to be positively related to the formation of positive emotions [59–61]. In short video marketing contexts, audiences develop emotional responses while processing multimodal information (e.g., images, music, and text) related to themes, products, or destinations. These emotional responses are subsequently associated with travel decision-making [62]. This process highlights the role of informational cues in eliciting positive emotions. In parallel, prior research has also demonstrated the importance of service-related factors in shaping positive emotional responses. For example, Kim and Lennon [63] pointed out that e-service quality is a prerequisite for users' online emotional experience; Rasheed [64] found that the high quality of online services predicts higher levels of consumers' positive emotions such as joy, arousal, and happiness. As an irrational motivator affecting visitor behavior, positive emotions are considered a decisive factor in predicting visitors' motivations and travel intention [60,65].

The integration of the SOR model and emotion appraisal theory in the previous analysis indicates that environmental stimuli are associated with cognitive evaluation, which is, in turn, related to emotional responses. In the tourism context, destination image is usually regarded as the overall cognitive evaluation formed after tourists integrate their judgments of destination environment and attributes [66]. A positive evaluation of destination image has been found to be positively associated with tourists' emotional experiences [67], evoking positive emotions such as surprise and happiness [68]. Based on this reasoning, in the context of GCTTAs, information quality and service quality are associated with positive emotions through destination image perception. Positive emotions have been shown to predict higher levels of approach motivation and travel intention. Based on the above analysis, the following hypotheses are proposed:

**H5:** Positive emotions mediate the relationship between the information quality of GCTTAs and audience travel intentions.

**H6:** Positive emotions mediate the relationship between the service quality of GCTTAs and audience travel intentions.

**H7:** Destination image perception and positive emotions play a sequential mediating role between information quality of GCTTAs and audience travel intentions.

**H8:** Destination image perception and positive emotions play a sequential mediating role between service quality of GCTTAs and audience travel intentions.

## 3. Research design

### 3.1. Variable measurement

All measurement scales were adopted from well-established prior studies. In the process of scale development, this study strictly followed cross-cultural research procedures and adopted a three-stage procedure to ensure the validity of the measurement instrument. First, based on mature international scales, localization adaptation was conducted. The original English questionnaire was translated and back-translated by the researchers, and wording adjustments were made to reduce semantic ambiguity. On this basis, two experts in tourism management were invited to provide suggestions and revisions. After several rounds of modification, the final questionnaire was confirmed. In the formal survey, information quality, service quality, destination image perception, positive emotions, and travel intention were measured using a seven-point Likert scale ranging from 1 strongly disagree to 7 strongly agree.

In the measurement model, information quality and service quality were specified as second-order latent constructs. Information quality of GCTTAs was conceptualized as a multidimensional construct consisting of content quality, usefulness quality, and presentation quality, each measured with three items. Service quality was measured using nine items across three dimensions: reliability, responsiveness, and assurance. The measurement items for information quality were adapted from Kim et al., Jiang et al., and DeLone and McLean [30,31,69]. For example, the item assessing whether

review information is easily understood in Jiang et al. [31] was revised to evaluate whether short videos posted by government cultural tourism TikTok accounts are easy to understand. The measurement items for service quality were adapted from Parasuraman et al., Zhang, and Kuo [18,41,70]. For example, the original item assessing whether the information provided by an official WeChat account is authoritative and trustworthy, proposed by Zhang [18], was adapted to assess whether GCTTAs are authoritative official sources of information. This adaptation improves contextual relevance and clarity of measurement while preserving the original construct meaning. The measurement of positive emotions was adapted from the scales developed by Qiu et al., Hosany et al. [68,71], including five items describing relaxed, pleasant, excited, interesting, and surprised feelings. Destination image perception was measured using three items adapted from Prayag et al., Baloglu and McCleary, Echtner and Ritchie [72–74]. For example, Prayag et al. [72] measured overall destination impression using semantic differential evaluations ranging from unfavourable to favourable. In this study, the evaluation was adapted to assess whether audiences perceive the destination as having a favorable overall image after watching the short videos. Travel intention was measured using the scale developed by Fu et al. [75], including three items assessing recommendation intention, visit intention, and positive word-of-mouth intention toward the destination.

### 3.2. Sample collection

This study adopted a non-probability sampling method. From mid September to early October 2023, electronic questionnaires were distributed through WeChat and TikTok platforms. The survey was administered through the Wenjuanxing platform, which was configured to prevent multiple submissions by blocking duplicate responses from the same login account, IP address, and device. The system automatically collected the questionnaires after submission. To improve sample quality and reduce invalid responses, several control measures were applied during data collection. First, a standardized explanation of GCTTAs was provided on the first page of the questionnaire. It clearly stated that these accounts are officially certified and operated by government culture and tourism departments at different administrative levels, usually with official verification marks on the TikTok platform. These accounts mainly publish destination promotion information and undertake public communication functions. Second, to ensure the relevance and validity of the sample, a screening question was included. The question asked whether the respondent had watched tourism short videos published by GCTTAs. Only respondents who had previously viewed or followed such content were included in the survey. Third, regarding recruitment procedures, the questionnaire was disseminated on WeChat primarily through tourism-related interest groups, with additional distribution facilitated via snowball sampling; on TikTok, participants were recruited through direct messaging, whereby users were randomly selected from those who had previously commented on short videos posted by GCTTAs and were invited to voluntarily participate in the survey. At last, based on testing about response time, questionnaires completed in less than 90 seconds were considered invalid and removed from the dataset to minimize careless responses.

This study collected data through an anonymous online survey. The survey did not involve sensitive or high-risk topics and was classified as minimal-risk research. All participants were informed of the research purpose, procedures, potential risks, and their rights before participation. For minors, parental or legal guardian consent was required prior to participation. Participation was entirely voluntary. Informed consent was obtained electronically at the beginning of the questionnaire. Only participants who clicked the Agree and Participate button were allowed to proceed, and submission of the completed questionnaire was regarded as consent to participate. No personally identifiable information was collected. The research protocol was approved by the institutional ethics committee and was conducted in accordance with relevant ethical guidelines.

A total of 424 questionnaires were collected. After screening based on whether respondents had watched GCTTAs and on response time less than 90 seconds, 406 valid questionnaires were retained, resulting in an effective response rate of 95.8 percent. In the sample, 44.1 percent were female. Generation Z respondents born between 1995 and 2009 accounted for 51.5 percent. A total of 230 respondents had a bachelor's degree, accounting for 56.7 percent, which was

the largest education group. In terms of occupation, company employees, students, and employees of public institutions accounted for 45.8 percent, 27.6 percent, and 15.8 percent, respectively. Regarding monthly income, 33.3 percent earned less than 3000 RMB, and 31.8 percent earned between 3001 and 5000 RMB. In terms of short video usage, 66 percent of respondents reported daily viewing time between 30 minutes and 2 hours, while 8.9 percent reported more than 2 hours per day. Another 25.1 percent reported less than 30 minutes per day. Overall, the sample mainly consisted of medium to high frequency users. This indicates that most respondents had stable and continuous usage experience with the TikTok platform, including its content ecosystem, interaction patterns, and information presentation format. Therefore, they were able to provide relatively reliable evaluations of the information and service performance of GCTTAs.

## 4. Data analysis and results

### 4.1. Reliability and validity analysis

SPSS 26.0 and AMOS 24.0 were used for data analysis. To evaluate the reliability and accuracy of the measurement instruments, several metrics were employed, including Cronbach's alpha, composite reliability (CR), and average variance extracted (AVE). As shown in Table 1, all standardized factor loadings of the measurement items are greater than 0.6, the CR values are above 0.8, and the AVE values are above 0.5. These results indicate that the latent constructs in this study demonstrate good convergent validity. Since the data were collected through self-reported questionnaires, Harman's single-factor test was conducted to examine the potential severity of common method bias. The results of principal component analysis show that the first principal component explains 38.928 percent of the total variance, which is below the threshold of 40 percent. This suggests that common method bias is not a serious concern in this study.

To examine the discriminant validity of the latent variables, confirmatory factor analysis (CFA) was conducted by comparing the theoretical five-factor model with several competing models. According to the recommended criteria summarized by Wu [76], a model demonstrates good fit when $\chi^2$/df falls between 1 and 3, GFI, NFI, IFI, and CFI are greater than 0.90, RMSEA is less than 0.08, and SRMR is less than 0.05. As shown in Table 2, except for GFI, all fit indices of the five-factor model (M1) satisfied these recommended thresholds. Although the GFI value of M1 is 0.885, which is slightly below the recommended threshold of 0.9, many researchers consider GFI values between 0.80 and 0.89 as indicating an acceptable model fit [77]. The $\chi^2$/df, RMSEA, GFI, NFI, IFI, and CFI values of M1 are all better than those of Models M2 to M6, indicating that the proposed model provides the best fit to the data. In particular, the model (M3) combining destination image perception and positive emotions into a single factor showed substantially poorer fit than the proposed five-factor model. This result suggests that organism states can be clearly distinguished into cognitive and emotional dimensions. This finding provides support for the proposed parallel mediation and serial mediation hypotheses. Overall, the result of CFA supports satisfactory discriminant validity among information quality, service quality, positive emotions, destination image perception, and travel intention.

Finally, to assess the potential influence of common method bias, an unmeasured latent method factor was incorporated into the baseline model to form Model M7. The changes in model fit indices between M1 and M7 were then examined. The results showed that $\Delta$GFI=0.015, $\Delta$RMSEA=0.003, $\Delta$NFI=0.012, $\Delta$IFI=0.009, and $\Delta$CFI=0.009. Following the model comparison guidelines proposed by Chen [78] for evaluating changes in fit indices, when the change in fit indices is less than 0.02, the improvement in model fit is considered insignificant. Therefore, the inclusion of the unmeasured latent method factor does not substantially improve model fit, suggesting that common method bias is unlikely to pose a serious threat to the findings.

### 4.2. Hypothesis testing

The structural equation model was conducted to test the hypothesized relationships, and the results are presented in Table 3. The path coefficient from information quality to travel intention is positive and significant ($\beta$=0.208, p<0.01);

**Table 1. Standardization factor loading, AVE, and CR values of variables.**

| Variable | Measurement questions | Std. factor loading | Cronbach's alpha | CR | AVE |
|---|---|---|---|---|---|
| **Information content quality** | Of sufficient depth. | 0.895 | 0.859 | 0.862 | 0.676 |
| | Specific. | 0.813 | | | |
| | Accurate. | 0.752 | | | |
| **Information utility quality** | Effective for planning a trip. | 0.777 | 0.832 | 0.842 | 0.641 |
| | Useful for planning a trip. | 0.756 | | | |
| | Helpful for planning a trip. | 0.864 | | | |
| **Information expression quality** | Easy to understand. | 0.893 | 0.829 | 0.840 | 0.640 |
| | Concise and appropriate. | 0.642 | | | |
| | Harmonious and consistent. | 0.844 | | | |
| **Responsiveness** | Respond promptly to online comments. | 0.818 | 0.809 | 0.816 | 0.601 |
| | Suggestions can receive timely feedback. | 0.867 | | | |
| | Have many interactive service features. | 0.619 | | | |
| **Assurance** | Trustworthy. | 0.85 | 0.825 | 0.826 | 0.614 |
| | Sufficient professional knowledge. | 0.784 | | | |
| | Friendly and courteous. | 0.71 | | | |
| **Reliability** | Updated promptly and stably. | 0.733 | 0.837 | 0.847 | 0.649 |
| | All kinds of services can be used. | 0.862 | | | |
| | Authoritative official information source. | 0.817 | | | |
| **Image perceived** | Attractive cultural or natural landscapes. | 0.77 | 0.870 | 0.872 | 0.694 |
| | Unique environmental ambiance. | 0.87 | | | |
| | Favorable overall image. | 0.856 | | | |
| **Positive emotions** | Relaxed. | 0.826 | 0.907 | 0.910 | 0.670 |
| | Pleasant. | 0.848 | | | |
| | Excited. | 0.83 | | | |
| | Interesting. | 0.701 | | | |
| | Surprising. | 0.876 | | | |
| **Travel intentions** | Recommend it to others. | 0.928 | 0.857 | 0.861 | 0.676 |
| | Intend to visit it in the future. | 0.801 | | | |
| | Say positive things about it to other people. | 0.725 | | | |

**Table 2. The results of the confirmatory factor analysis.**

| Model | Included factors | χ2 | df | χ2/df | GFI | RMSEA | NFI | IFI | CFI | SRMR |
|---|---|---|---|---|---|---|---|---|---|---|
| **M1** | IQ、SQ、PE、IP、TI | 738.561 | 361 | 2.046 | 0.885 | 0.051 | 0.906 | 0.950 | 0.949 | 0.049 |
| **M2** | IQ＋SQ、PE、IP、TI | 951.952 | 365 | 2.608 | 0.850 | 0.063 | 0.879 | 0.922 | 0.921 | 0.070 |
| **M3** | IQ, SQ, PE＋IP, TI | 1230.200 | 365 | 3.370 | 0.808 | 0.077 | 0.843 | 0.884 | 0.884 | 0.071 |
| **M4** | IQ＋SQ＋PE, IP, TI | 1306.112 | 368 | 3.549 | 0.793 | 0.079 | 0.834 | 0.875 | 0.874 | 0.101 |
| **M5** | IQ＋SQ＋PE＋IP, TI | 1678.793 | 370 | 4.537 | 0.740 | 0.093 | 0.786 | 0.825 | 0.824 | 0.094 |
| **M6** | IQ＋SQ＋PE＋IP＋TI | 1875.556 | 371 | 5.055 | 0.720 | 0.100 | 0.761 | 0.799 | 0.798 | 0.092 |
| **M7** | ULMF, IQ, SQ, PE, IP, TI | 646.149 | 335 | 1.929 | 0.900 | 0.048 | 0.918 | 0.959 | 0.958 | 0.048 |

*Note:* IQ:information quality; SQ: service quality; PE: positive emotions; IP: image perception; TI: travel intentions; ULMF: Unmeasured latent method factor.

**Table 3. Structural model testing.**

| Path | Point estimation | Product of coefficient | | Bias-Corrected 95%CI | | Percentile 95%CI | |
|---|---|---|---|---|---|---|---|
| | | SE | Z | Lower | Upper | Lower | Upper |
| **Direct effect** | | | | | | | |
| IQ-->TI | 0.208 | 0.066 | 3.152 | 0.083 | 0.349 | 0.081 | 0.346 |
| SQ-->TI | 0.025 | 0.068 | 0.368 | −0.104 | 0.164 | −0.107 | 0.162 |
| **Indirect effect** | | | | | | | |
| IQ-->IP-->TI | 0.133 | 0.033 | 4.030 | 0.080 | 0.212 | 0.076 | 0.205 |
| SQ-->IP-->TI | 0.160 | 0.051 | 3.137 | 0.079 | 0.286 | 0.074 | 0.278 |
| IQ-->PE-->TI | 0.073 | 0.025 | 2.920 | 0.035 | 0.137 | 0.030 | 0.128 |
| SQ-->PE-->TI | 0.041 | 0.024 | 1.708 | 0.007 | 0.107 | 0.005 | 0.100 |
| IQ-->IP-->PE-->TI | 0.018 | 0.009 | 2.000 | 0.005 | 0.044 | 0.004 | 0.040 |
| SQ-->IP-->PE-->TI | 0.022 | 0.011 | 2.000 | 0.007 | 0.053 | 0.006 | 0.048 |
| **Total effect** | | | | | | | |
| IQ-->TI | 0.432 | 0.069 | 6.261 | 0.303 | 0.575 | 0.302 | 0.575 |
| SQ-->TI | 0.248 | 0.083 | 2.988 | 0.102 | 0.426 | 0.103 | 0.427 |

*Note:* Z=Point estimation/ Standard Error. IQ:information quality; SQ: service quanlity; PE:positive emotions; IP:image perceived; TI:travel intentions.

therefore, H1 is supported. The path coefficient from service quality to travel intention is not significant (β=0.025, p>0.05); therefore, H2 is not supported.

Bootstrapping analyses were performed to assess indirect associations. The results indicate that information quality was significantly indirectly associated with travel intention through destination image (β=0.133, p<0.001) and through positive emotions (β=0.073, p<0.01). A significant sequential indirect association through destination image and positive emotions was also observed (β=0.018, p<0.05). The bias-corrected and percentile 95% confidence intervals for these indirect paths did not include zero. Therefore, H3, H5, and H7 are supported. Both the direct and indirect effects of information quality on travel intention were significant and positive, indicating a pattern of complementary mediation [79]. Similarly, service quality demonstrated significant indirect associations with travel intention through destination image (β=0.160, p<0.01) and positive emotions (β=0.041, p<0.05), as well as a significant sequential indirect association via destination image and positive emotions (β=0.022, p<0.05). The bias-corrected and percentile corresponding 95% confidence intervals excluded zero. Therefore, H4, H6, and H8 are supported. As the direct association between service quality and travel intention was not significant, this pattern corresponds to indirect-only mediation [79].

### 4.3. Robustness tests

Multi-group structural equation modeling (MGSEM) is widely used to examine heterogeneity across different populations. This approach enables the simultaneous analysis of multiple samples to assess whether the conceptual model proposed in the study demonstrates consistency or variation across different groups [80]. The core of multi-group structural equation modeling lies in testing measurement and structural invariance by imposing a series of increasingly restrictive parameter constraints and comparing nested models across groups. Specifically, the analysis typically begins with an unconstrained baseline model, in which no equality constraints are imposed across groups. Subsequently, parameter constraints are introduced step by step, and each constrained model is compared with the preceding model. Changes in model fit indices (e.g., CFI and RMSEA), as well as their differences (e.g., ΔCFI), are examined to determine whether the imposition of constraints leads to a significant deterioration in model fit, thereby indicating whether invariance holds across groups. The commonly examined nested models include the unconstrained model, measurement weights model, structural weights

model, structural covariances model, structural residuals model, and measurement residuals model. These models collectively represent a hierarchical testing process from measurement invariance to structural invariance.

Gender and age are important segmentation variables in marketing research, as consumers of different demographic groups may exhibit distinct patterns in purchase behavior, brand preference, and product usage. Accordingly, this study conducted multi-group analyses based on gender and age. Specifically, the total sample was divided into male (N = 227) and female (N = 179) groups. For age classification, the original sample consisted of respondents born before 1980 (N = 36), Millennials (1980–1994, N = 154), Generation Z (1995–2009, N = 209), and those born after 2010 (N = 7). Given that multi-group structural equation modeling requires a reasonable balance in sample size, with the ratio between groups not exceeding 2:1 (Hair et al., 2018), the age groups were further consolidated. Respondents born in or before 1994 (N = 190) were categorized as the older group, while those born in or after 1995 (N = 216) were classified as the younger group, thus satisfying the sample size requirements for multi-group analysis.

In addition, this study employed daily TikTok usage duration as a key indicator of user engagement and adopted one hour as the cutoff point to distinguish usage frequency. Prior research suggests that limiting social media use to approximately 60 minutes per day represents a meaningful behavioral threshold, as exceeding this level is associated with lower well-being and more negative psychological outcomes [81]. Accordingly, respondents who reported using TikTok for less than one hour per day (within 30 minutes, N = 102; 30 minutes to 1 hour, N = 122) were classified as the low-frequency group (N = 224), whereas those who used TikTok for more than one hour per day (1–2 hours, N = 146; more than 2 hours, N = 36) were categorized as the high-frequency group (N = 182).

To assess the robustness of the proposed model across different groups, multi-group invariance tests were conducted based on gender, age, and usage frequency, and the results are presented in Table 4. The unconstrained models exhibit acceptable fit across all groups (CFI > 0.90, RMSEA < 0.08), supporting configural invariance. As increasingly restrictive constraints were imposed, changes in model fit remained minimal, and the chi-square difference tests for most nested models were non-significant (p > 0.05). Although the chi-square difference test is significant in some cases (e.g., the

**Table 4. Multi-group invariance test results across gender, age, and usage frequency.**

| | Model | χ2/df | CFI | RMSEA | Δdf | Δχ2 | p | ΔCFI |
|---|---|---|---|---|---|---|---|---|
| **Gender** | Unconstrained | 1.669 | 0.937 | 0.041 | — | — | — | — |
| | Measurement weights | 1.651 | 0.936 | 0.040 | 20 | 20.545 | 0.424 | −0.001 |
| | Structural weights | 1.673 | 0.933 | 0.041 | 13 | 38.012 | 0.000 | −0.003 |
| | Structural covariances | 1.671 | 0.933 | 0.041 | 3 | 3.375 | 0.337 | 0.000 |
| | Structural residuals | 1.664 | 0.933 | 0.041 | 9 | 9.420 | 0.399 | 0.000 |
| | Measurement residuals | 1.670 | 0.930 | 0.041 | 29 | 52.964 | 0.004 | −0.003 |
| **Age** | Unconstrained | 1.526 | 0.948 | 0.036 | — | — | — | — |
| | Measurement weights | 1.519 | 0.947 | 0.036 | 20 | 25.686 | 0.176 | −0.001 |
| | Structural weights | 1.513 | 0.948 | 0.036 | 4 | 1.344 | 0.854 | 0.001 |
| | Structural covariances | 1.513 | 0.947 | 0.036 | 3 | 4.482 | 0.214 | −0.001 |
| | Structural residuals | 1.511 | 0.947 | 0.036 | 9 | 12.161 | 0.204 | 0.000 |
| | Measurement residuals | 1.512 | 0.945 | 0.036 | 29 | 44.963 | 0.030 | −0.002 |
| **Usage Frequency** | Unconstrained | 1.584 | 0.943 | 0.038 | — | — | — | — |
| | Measurement weights | 1.560 | 0.944 | 0.037 | 4 | 5.198 | 0.268 | 0.001 |
| | Structural weights | 1.559 | 0.944 | 0.037 | 3 | 2.105 | 0.551 | 0.000 |
| | Structural covariances | 1.555 | 0.944 | 0.037 | 9 | 9.141 | 0.424 | 0.000 |
| | Structural residuals | 1.549 | 0.944 | 0.037 | 29 | 45.17 | 0.028 | 0.000 |
| | Measurement residuals | 1.549 | 0.942 | 0.037 | 4 | 5.198 | 0.268 | −0.002 |

structural weights model for gender and the measurement residuals models for gender and age), the corresponding ΔCFI values remain well below the 0.01 threshold, supporting the invariance assumption [82].

Overall, the multi-group invariance test results demonstrate that the proposed model is invariant across gender, age, and usage frequency groups, suggesting no meaningful group differences and confirming the robustness of the findings.

## 5. Conclusions and implications

### 5.1. Conclusions

This study finds that in the context of GCTTAs, information stimuli and service stimuli relate to travel intention through different structural patterns. This difference can be further explained from the perspective of information processing. The HSM proposes that individuals may process information through a systematic route by carefully analyzing message content, or through a heuristic route by relying on simple cues to make quick judgments [83].

In the context of GCTTAs, the main communicators are staff from government culture and tourism departments. This specific institutional identity provides them with inherent authority and credibility, which enables government short videos to transmit information more effectively [84]. The heuristic processing route suggests that individuals can make rapid judgments based on simple informational cues, such as relying on the perceived reliability of expert opinions [85]. When individuals are exposed to information from highly credible sources, their perceived risk and decision uncertainty are reduced. As a result, they are more likely to adopt a heuristic processing strategy and make quick evaluations [86,87]. This shortened cognitive path means that information quality can directly predict audience travel intention. Hu et al. [88] pointed out that practical information, such as completeness and relevance, is more likely to trigger tourists' systematic processing than recreational information. Therefore, when information is complex or involves personal interests, individuals tend to engage in systematic processing, carefully analyzing the content and logic to weigh pros and cons before making judgments. Thus, high-quality information provided by GCTTAs is associated with audience travel intentions through multiple cognitive pathways.

Compared with information quality, the service quality of GCTTAs is not significantly directly associated with audience travel intentions. This finding may be attributed to the evaluative nature of service quality [89]. Based on the HSM, service quality is a comprehensive evaluation formed through ongoing interactions and typically requires greater cognitive effort and systematic processing [90]. In the context of GCTTAs, service quality depends on ongoing interactions and relationship maintenance. As such, limited short-term interactions make it difficult for users to form stable evaluations, reducing the likelihood that service quality serves as an immediate cue directly associated with travel intentions. However, as interactions between audiences and GCTTAs accumulate over time and trust develops, service quality is associated with more favorable destination image perception and positive emotions, and is indirectly associated with users' travel intentions through these pathways [91]. This suggests that service quality is essentially a relationship-oriented stimulus, and its association with audience travel intentions operates primarily through cognitive and emotional mediating mechanisms rather than through direct links. This mechanism also offers a theoretical explanation for the sequential mediation from destination image to positive emotions identified in this study. Furthermore, from the perspective of media context, the algorithm-driven recommendation mechanisms and fast-paced content consumption on short-video platforms reinforce audiences' reliance on heuristic processing, creating a decision-making environment characterized by low cognitive effort and rapid judgments. In such contexts, audiences tend to rely more on intuitive informational cues and emotional experiences, while engaging less in systematic evaluation of service interactions. As service quality inherently depends on systematic processing, a mismatch arises between its underlying processing route and the heuristic-dominated decision environment, thereby weakening its immediate impact on travel intentions.

Destination image perception and positive emotions are statistically associated with the relationships between information quality, service quality, and audience travel intentions in the context of GCTTAs. This finding is consistent with

previous studies [55,91–93]. The present study provides additional empirical support for this mediation pattern within the context of GCTTAs. This suggests that, in the context of social media marketing by government culture and tourism departments, audiences' overall cognitive evaluation of a destination and their emotional experience remain important psychological bridges in the formation of travel intention.

In addition, this study not only provides evidence of mediation patterns but also further explores the associations among organism variables. Different from prior research that emphasizes the pathway in which emotion is associated with destination image, which in turn is related to travel intention [94,95], this study suggests an alternative directional association. Specifically, in the communication context of GCTTAs, destination image perception is positively associated with positive emotions, which is further linked to travel intention. This finding is consistent with emotion appraisal theory, which posits that emotions arise from individuals' cognitive appraisal of external stimuli [23]. This finding provides additional empirical support for the applicability of the emotion appraisal theory in the context of social media marketing by government culture and tourism departments.

### 5.2. Managerial implications

First, the formation of travel intentions is strongly associated with information quality. Short videos that deliver richer information and stimulate multiple senses are considered more persuasive in shaping customer attitudes and behaviors [96]. Therefore, government culture and tourism departments should not only provide decision-relevant information in their content production, but also intentionally incorporate image-enhancing elements and emotional cues. At the operational level, in the first three seconds of a video, distinctive visual scenes and structured titles can be used to quickly present the unique cultural symbols of the destination. In the main content, information density may be enhanced by clearly presenting key decision information, such as featured experiences, the best travel period, and scenic spot ticket policies. At the same time, by integrating authoritative information sources, real scene displays, and narrative storytelling of personal experiences, an immersive atmosphere can be created to stimulate positive emotions such as pleasure and anticipation. By integrating decision support, image reinforcement, and emotional expression within the same content framework, GCTTAs can simultaneously strengthen the direct association of information quality with travel intention and its indirect associations through cognitive and emotional pathways.

Second, the core value of service management is reflected in its role in influencing destination image and positive emotions. Social media is not only a channel for information release, but also a platform for user interaction. Therefore, government culture and tourism departments should adopt a relationship-oriented service perspective and enhance their responsiveness to users' information needs through the continuous provision of clear, reliable, and targeted decision-support information, thereby improving service professionalism and interaction quality. They can also monitor and analyze destination-related information published by suppliers, intermediaries, residents, media, and tourists, and respond through clear intervention strategies [97]. This process strengthens the link between perceived service professionalism and destination reliability, thereby contributing to the accumulation of destination brand equity over time. In addition, by using the service functions of the TikTok platform, GCTTAs can embed an intelligent service window in the short video interface and provide real-time consultation through AI customer service. They can also conduct regular live streaming sessions to answer questions and enhance authenticity and sense of presence through face-to-face communication. Continuous interactive experiences help lower perceived decision uncertainty and reinforce positive emotions, both of which are closely linked to travel intention. Furthermore, consumers tend to prefer products and brands from companies that allow them to submit feedback [98]. Therefore, government culture and tourism departments can enhance audiences' sense of participation and belonging by collecting user feedback or encouraging user co-creation. In this way, the service process itself functions as an important source of emotional value in destination communication.

Third, this study finds that destination image perception precedes positive emotions, which suggests that short video content should follow a psychological path from cognitive construction to emotional reinforcement. In practice, GCTTAs

can continuously strengthen the core image positioning through thematic series content. For example, structured content can be developed around dimensions such as regional culture, natural landscapes, or local lifestyle. In this process, highly recognizable regional cultural IP elements, such as landmark architecture, traditional customs, distinctive dialects, or representative symbols, can be embedded to enhance image distinctiveness and cognitive clarity. Emotional appeal can be enhanced through techniques such as music rhythm, varied camera work, and storytelling of personal experiences. These techniques contribute to an immersive experience and a sense of involvement, which are linked to the emergence of positive emotions following cognitive evaluation.

### 5.3. Limitations and future research

This study has several limitations. First, this study is based on cross-sectional survey data, with all variables measured at a single point in time; therefore, strict causal inferences cannot be established. Although the structural model was developed based on well-established theoretical frameworks and the relationships among variables were tested using structural equation modeling, the findings should be interpreted as statistical associations rather than definitive causal effects. Future research could employ longitudinal designs or experimental methods to more rigorously examine the causal relationships among these variables.

Second, the model proposed in this study is based on the characteristics of GCTTAs. Therefore, the applicability of the findings to other social media platforms or to culture and tourism communication contexts in different countries still needs further examination. Future research can conduct comparative analyses across different types of social media platforms or carry out cross-cultural studies to explore the similarities and differences of government culture and tourism marketing mechanisms under different platform features and cultural backgrounds.

Finally, this study adopted a linear structural equation model to analyze the relationships among variables. However, the short video communication environment is characterized by high interactivity and algorithmic feedback. Audience destination image perception and emotional responses may exhibit nonlinear patterns or threshold effects. Future research could integrate machine learning approaches or complex systems analysis to better understand how social media marketing by government culture and tourism departments is associated with audience travel intentions.

## Supporting information

**S1 File. Survey questionnaire.**
(PDF)

**S2 File. Dataset used for analysis.**
(XLS)

## Author contributions

**Conceptualization:** Bin Wen.

**Data curation:** Yuan Sun.

**Formal analysis:** Yuan Sun.

**Funding acquisition:** Yuan Sun.

**Investigation:** Yuan Sun.

**Methodology:** Yuan Sun.

**Writing – original draft:** Yuan Sun.

**Writing – review & editing:** Bin Wen.

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
