## [Decision Letter · Decision Letter 0]

30 Jan 2026

PONE-D-25-66483Investigating the Effects of Information Quality and Service Quality of Government Culture and Tourism Douyin accounts on Audience Travel Behavioral IntentionsPLOS One

Dear Dr. Sun,

Thank you for submitting your manuscript to PLOS ONE. After careful consideration, we feel that it has merit but does not fully meet PLOS ONE’s publication criteria as it currently stands. Therefore, we invite you to submit a revised version of the manuscript that addresses the points raised during the review process.

We look forward to receiving your revised manuscript.

Kind regards,

Hai-Tao Yu, ph.D.

Academic Editor

PLOS One

**Journal Requirements:**

“This study is funded by the "Anhui Provincial Philosophy and Social Science Planning Project". Project Approval Number: AHSKYQ2023D015.”

5. We note that your Data Availability Statement is currently as follows:

“All relevant data are within the manuscript and its Supporting Information files.”

6. We are unable to open your Supporting Information file [ Questionnaire survey data.sav] Please kindly revise as necessary and re-upload.

Reviewers' comments:

Reviewer's Responses to Questions

**Comments to the Author**

1. Is the manuscript technically sound, and do the data support the conclusions?

Reviewer #1: Partly

Reviewer #2: Yes

Reviewer #3: Yes

Reviewer #4: Yes

2. Has the statistical analysis been performed appropriately and rigorously? 

Reviewer #1: Yes

Reviewer #2: Yes

Reviewer #3: Yes

Reviewer #4: Yes

3. Have the authors made all data underlying the findings in their manuscript fully available?

Reviewer #1: Yes

Reviewer #2: Yes

Reviewer #3: Yes

Reviewer #4: No

4. Is the manuscript presented in an intelligible fashion and written in standard English?

Reviewer #1: Yes

Reviewer #2: No

Reviewer #3: No

Reviewer #4: Yes

5. Review Comments to the Author

Reviewer #1: This study applies the SOR model and emotion appraisal theory to examine how information quality and service quality of government culture and tourism Douyin accounts influence travel behavioral intention through positive emotions and destination image perception. The topic is timely, the structure is sound, and the methodological approach is generally appropriate. However, the manuscript requires major revisions to strengthen its theoretical contribution, literature engagement, methodological transparency, and depth of discussion.

1. The introduction does not sufficiently highlight the theoretical innovation of the study. While the introduction outlines the operational complexity of government accounts and mentions the dual attributes of policy advocacy and public service, it fails to clearly articulate how this research advances existing theoretical frameworks. For instance, it states that government accounts must "balance policy advocacy and public service responsibilities," but does not explicitly explain how integrating service quality into the SOR model—typically focused on information or system quality—creates a novel "dual-driver" framework for public sector communication. The concluding paragraph of the introduction should be revised to sharply contrast this study's integrated model (information + service quality) with prior SOR applications in commercial or UGC contexts, thereby underscoring its distinct theoretical contribution to understanding government social media effects.

2. The literature review lacks critical synthesis and reads more like a summary list, failing to establish a clear dialogue with existing research. The review categorizes short-video marketing into three paradigms but does not critically analyze their limitations when applied to government-operated accounts. For example, it notes that most studies focus on information quality while overlooking service quality but does not engage with relevant public administration or e-government literature discussing the service role of official social media (e.g., Zhang et al., 2017 on government WeChat service quality). To strengthen the argument, the review should explicitly position itself against prior work, explaining why the service dimension is especially salient and under-theorized for government tourism accounts, and how this study's combined focus addresses a defined gap.

3. The methods section is described too briefly, omitting key details necessary for replicability and assessment of validity. The adaptation process of measurement scales is only vaguely described as involving "translation and back-translation" and expert review. The manuscript should provide specific examples of how items were modified to fit the context of government Douyin accounts (e.g., original scale item vs. adapted item). Furthermore, the sampling procedure states that questionnaires were distributed "via WeChat and Douyin" but lacks details on the sampling frame, recruitment channels, or measures to prevent duplicate or inattentive responses (e.g., attention checks). Additionally, while Harman's single-factor test is mentioned, the result (38.93%) is not reported in the main text or tables. These omissions hinder the evaluation of the study's rigor.

4. The discussion of results is relatively superficial, primarily restating findings without deeply interpreting their theoretical significance or contextualizing them within the broader literature. For instance, the key finding that service quality exerts only an indirect effect (full mediation) while information quality has both direct and indirect effects is noted but not sufficiently explained. The discussion attributes this to "fragmented browsing" habits but does not connect it to theoretical mechanisms such as the heuristic processing of authoritative information versus the systematic evaluation required for service interactions. Moreover, the contrast with commercial tourism accounts—where service quality may have a direct impact—is hinted at but not substantiated with references to comparable studies. The discussion should explicitly link each major result back to the SOR and emotion appraisal frameworks, explore alternative explanations, and compare findings with related work to clarify the unique behavioral pathways in government social media contexts.

Reviewer #2: 1) Clarify theoretical contribution beyond “applying SOR in a new context”

The manuscript uses a well-established SOR structure (stimuli → organism → response). At present, the theoretical novelty is not sufficiently articulated. Please explicitly state what new theoretical insight emerges from studying government Douyin accounts rather than commercial/UGC contexts. Suggested actions:

Add a dedicated subsection (end of Introduction or start of Discussion) titled “Theoretical Contributions” with 2–3 specific points, e.g.:

how public-sector social media changes the role of service quality versus information quality.

why service quality effects are fully mediated in this context.

what this implies for extending SOR or public communication theory.

2) Improve conceptual distinctiveness: information quality vs service quality

On short-video platforms, boundaries between content features and interactive/service features can be blurred. Some service-quality items (e.g., “many interactive service features”) may overlap with perceptions of platform/content design. Please:

Provide a sharper conceptual rationale for treating information quality as content-based stimulus and service quality as interaction/response-based stimulus, and explain why these remain distinct in government TikTok  contexts.

Consider adding additional evidence of distinctiveness:

alternative discriminant validity checks (e.g., HTMT), or report cross-loadings / confirmatory factor comparisons.

3) Common method bias: Harman’s single-factor test is insufficient

The paper relies on Harman’s single-factor test, which is widely considered a weak diagnostic for CMB in self-reported cross-sectional designs. Please include at least one more rigorous approach, for example:

Common latent factor (CLF) technique in CFA/SEM and report whether substantive paths materially change, or Marker-variable approach, or Unmeasured latent method factor model.

Also clarify procedural remedies (e.g., anonymity, item order, instructions) beyond reporting statistics.

4) Robustness and alternative models

To strengthen confidence that the proposed mediation structure is not merely one plausible specification, please add at least one robustness/alternative-model test:

Compare the hypothesized model against reasonable alternative models, such as:

reversing organism variables (image → emotion vs emotion → image),

including a direct path from service quality to behavioral intentions but testing nested model comparisons, collapsing the two mediators into one (if justified) to show your model fits better.

Optional but valuable: multi-group SEM (e.g., Gen Z vs non-Gen Z, gender), given the demographic structure reported. Even if exploratory, it improves interpretability and external validity.

5) Causal language should be toned down

The dataset is cross-sectional and self-reported. Please avoid strong causal wording such as “influences/drive/enhances” unless you clearly frame them as associations. Suggested revisions:

Replace causal verbs with “is positively associated with / relates to / predicts (in the statistical sense)”.

Strengthen the limitations section explicitly: causal inference is limited and longitudinal/behavioral data are needed.

6) Language and style require professional editing

There are recurrent grammar issues, awkward phrasing, and some “Chinglish” expressions (e.g., “Government accounts have service attribute”). The manuscript would benefit from professional English editing. This is not cosmetic: readability affects reviewability and perceived rigor.

Minor Comments

Hypotheses numbering/logic: H1 and H3 appear overlapping (information quality → behavioral intentions). Please check redundancy and ensure each hypothesis is unique and consistent with the model narrative.

Measurement detail: provide the full item wording in a supplement (or an appendix) and specify how items were adapted (what modifications were made). This improves reproducibility.

Model fit reporting: consider reporting additional indices commonly requested (e.g., SRMR) and note thresholds with citations.

Summary

The manuscript addresses a relevant topic and reports plausible findings, but it requires major revisions to strengthen theoretical contribution, ensure construct distinctiveness, and substantially polish English writing.

Reviewer #3: The manuscript explores the use of Douyin by government agencies for tourism promotion. The topic is timely and holds clear practical relevance. However, there are still several concerns that need to be addressed:

Introduction:

1. For publication in an international journal, the use of proper nouns must be precise. The authors should verify whether "Douyin" is the standard accepted English terminology in current academic literature, or if it requires clarification (e.g., explicitly defining it as the Chinese version of TikTok) to ensure clarity for an international readership.

2. The authors assert that “the operational logic of government culture and tourism accounts is more complex” due to the need to balance policy advocacy and public service with user engagement. While this distinction is intuitive and pivotal to the study, the statement is currently presented without sufficient theoretical grounding.

4. The authors postulate that government culture and tourism accounts face a complex dual mandate involving 'policy advocacy.' However, this claim must be aligned with empirical reality. Based on the examples provided in the manuscript, the content appears to function primarily as generic destination promotion, indistinguishable from standard user-generated content. The authors need to clarify whether policy advocacy is genuinely embedded in the analyzed videos. If these accounts merely replicate the promotional style of the general public, the authors must explicitly delineate what constitutes the unique difference between government and public accounts to justify the theoretical necessity of focusing specifically on this group.

5. There is a noticeable lack of logical cohesion between the second and third paragraphs of the Introduction.

6. The study adopts the S-O-R framework and Appraisal Theory of Emotion as its theoretical foundation. However, the rationale for selecting these specific lenses remains unclear. The authors have not sufficiently explained why these theories are the most appropriate tools for this specific research context, nor have they articulated their unique value in explaining the observed phenomena.

7. The Introduction currently lacks a deep exploration of the study's theoretical and practical implications. The authors should elaborate on these aspects to clearly demonstrate how the research advances existing theoretical boundaries and provides substantive, actionable insights for destination management, rather than relying on broad generalizations.

Literature Review and Research Hypotheses:

1. The Literature Review section is currently too descriptive and lacks critical depth. Rather than merely cataloging previous studies, the authors should engage in a more analytical synthesis of the existing body of knowledge.

2. The exposition of emotional appraisal theory is currently inadequate. The manuscript fails to provide a clear, authoritative definition of the theory and lacks a comprehensive review of its recent research progress or evolutionary trajectory within the domain. To strengthen the theoretical framework, the authors should explicitly define the core tenets of the theory and synthesize relevant contemporary studies to demonstrate its current standing and applicability to this research.

3. The integration of the S-O-R framework and emotional appraisal theory within the hypothesis development section remains ambiguous. Specifically, it is unclear how these theoretical lenses are operationalized to deduce the proposed relationships.

Research Design

1. It is crucial to understand how the concept of 'government culture and tourism Douyin accounts' was introduced to the study participants. The manuscript should explicitly detail the instructions, definitions, or visual examples provided to respondents to ensure they correctly identified the specific type of account under investigation. The authors need to clarify this procedure in the methodology section to rule out potential confusion with general travel influencers or commercial accounts.

2. The reporting of sample characteristics regarding usage duration is currently incomplete. While it is noted that 66% of respondents spend 30 minutes to 2 hours daily, the distribution of the remaining 34% is not disclosed. Given the study's focus on Douyin, were specific screening criteria applied to ensure all respondents are active users? The authors must clarify the usage patterns of the remaining participants and justify whether the entire sample possesses sufficient platform experience to provide valid responses.

Conclusions and Implications

1. The Discussion section currently functions more as a summary of the results rather than a platform for theoretical synthesis. The authors predominantly reiterate their findings without explicitly articulating the study's theoretical contributions. Crucially, there is a lack of critical dialogue with the existing literature; the authors need to discuss how their findings either corroborate, contradict, or extend previous studies to clearly delineate the study's unique value to the field.

Limitations and Future Research

1. Several of the acknowledged limitations are remediable within the current research scope.

Reviewer #4: Dear Authors,

Thank you for submitting your work to PLOS One. I have had the opportunity to revise your paper, “Investigating the Effects of Information Quality and Service Quality of Government Culture and Tourism Douyin accounts on Audience Travel Behavioral Intentions.” This manuscript represents a methodologically sound and theoretically grounded contribution to the literature on government social media, tourism communication, and short-video platforms. I find it generally well written and potentially interesting to readers.

The manuscript is generally clear, well-structured, and intelligible, and the argument progresses logically from theory to hypotheses, methods, results, and implications. The literature review is comprehensive, and the discussion appropriately links findings back to theory and practice. The manuscript would further benefit from language polishing and copyediting (e.g., occasional redundancy and overly long sentences, particularly in the Introduction and Conclusions; grammatical errors and phrasing; inconsistent use of terminology such as “image perceived” vs. “image perception”; minor formatting and spacing issues in tables and hypothesis labeling). These issues do not undermine the scientific content but should be addressed to improve clarity and readability.

Overall, the manuscript presents a technically sound empirical study grounded in well-established theoretical frameworks (SOR model and emotion appraisal theory). The research questions are clearly articulated, the conceptual model is logically derived from literature, and the hypothesized relationships are theoretically justified. The study relies on survey data from 406 respondents, which is an adequate sample size for SEM. Measurement scales are adapted from prior validated studies, and the authors follow appropriate procedures for scale adaptation, including translation/back-translation and expert review. Reliability and validity assessments meet commonly accepted thresholds, supporting the adequacy of the measurement model. The SEM results are coherent and largely consistent with the proposed hypotheses. The distinction between the effects of information quality (direct and indirect) and service quality (fully mediated) is empirically supported and interpreted in a theoretically meaningful way. The conclusions regarding the “content-first” logic of government Douyin accounts are reasonable given the data. The outcome variable is behavioral intention rather than actual behavior, a limitation the authors appropriately note. The only theoretical issue worth improving is, in my opinion, a potential redundancy and overlap between information quality, destination image perception and positive emotions. For example, high-quality tourism videos by definition shape destination image, while emotional responses are tightly embedded in image formation. While discriminant validity statistics pass, this does not fully resolve conceptual overlap (information quality predicts image, which predicts intention, but image is partly constructed through information quality). Perhaps you could clarify temporal or cognitive ordering? (e.g., cognition  affect  image  intention). Overall, the conclusions are appropriately drawn from the data and do not overreach excessively the empirical evidence, but the findings should be interpreted within the acknowledged methodological constraints (i.e., non-probability sampling, cross-sectional design, self-reported measures). That said, I would like to call the attention tosomewhat overconfident causal interpretation and suggest reframing conclusions using associational language (using e.g., expressions such as "is associated with”, “predicts”, or “is linked to”) and acknowledging causal limitations more explicitly.

Statistical analysis is generally rigorous and appropriate for the research objectives. However, I believe the manuscript would benefit from a clearer explanation of how higher-order constructs (e.g., information quality and service quality) were modeled, that is, whether as second-order factors or aggregated dimensions. Also, some hypotheses (e.g., H3) appear conceptually redundant or mislabeled, which may reflect minor inconsistencies in hypothesis numbering rather than analytical flaws. Despite these minor issues, the analytical strategy is sound, competently executed, and aligned with current standards in tourism and social media research

The authors state that all relevant data are within the manuscript and its supporting information files. This is formally compliant with the PLOS Data Policy. However, based on the manuscript text alone, it is not entirely clear whether item-level response data (i.e., the data underlying means, variances, and SEM analyses) are fully accessible. If privacy or ethical constraints apply, these should be stated explicitly. Clarifying this point would strengthen compliance with PLOS ONE’s open data requirements.

6. PLOS authors have the option to publish the peer review history of their article (what does this mean?). If published, this will include your full peer review and any attached files.

Reviewer #1: No

Reviewer #2: **Yes:** Yuan Li

Reviewer #3: No

Reviewer #4: **Yes:** Nina Szczygiel

---

## [Author Response · Author response to Decision Letter 1]

10 Mar 2026

Response Letter

Dear editors and reviewers,

We are very grateful for your constructive comments and suggestions for our manuscript entitled "Investigating the effects of information quality and service quality of government culture and tourism TikTok accounts on audience travel behavioral intentions" (ID:PONE-D-25-66483). Your comments are very valuable and helpful for improving our manuscript. In the following, we provide point-by-point responses to all comments.

We have carefully revised the manuscript to ensure that all changes are clearly presented, and we hope that the revised manuscript can meets the requirements for publication. The main revisions in the new manuscript are:

1.In the introduction, we added a review of the SOR model and emotion appraisal theory to construct the theoretical framework for this study. The introduction also clearly outlines the theoretical contributions of the study.

2.In the literature review and hypothesis development sections, we reorganized existing studies and proposed the hypotheses based on a critical review of prior research. To maintain clarity, unrelated hypotheses were removed, keeping only the direct effects, parallel mediation effects, and sequential mediation effects between information quality, service quality, and audience travel intentions.

3.In the variable measurement section, We added a comparison between original and adapted measurement items, using selected examples to demonstrate how the scales were adjusted to fit the research context.

4.In the sample collection section, we provided additional details about the survey procedures, including the sampling frame, recruitment channels, and measures to prevent duplicate or careless responses. In accordance with standard academic practice, an ethics statement has been incorporated into the manuscript.

5.In the data analysis section, we conducted confirmatory factor analysis with competing models to assess construct validity. In addition, we incorporated an unmeasured latent method factor into the structural model into the structural model to assess the potential impact of common method bias.

6.We have substantially revised the Discussion to strengthen the theoretical interpretation. The revision integrates the heuristic–systematic model and emotion appraisal theory, clarifies the distinct roles of information quality and service quality, and shows that destination image perception occurs before and is linked to positive emotions, highlighting how cognitive evaluation is linked to emotional responses and strengthening the study’s theoretical insights.

7.The limitations section has been revised to acknowledge the cross-sectional design, context dependence, and linear modeling assumptions, and we suggest that future research explore longitudinal, cross-platform, and nonlinear approaches to better understand user responses in social media environments.

We would also like to note that, following the substantial revisions made to strengthen the theoretical framework and clarify the study’s contributions, we have revised the manuscript title to better reflect the updated focus and theoretical positioning of the study.The revised title is: “Understanding travel intention formation in government culture and tourism TikTok accounts: An integration of the SOR model and emotion appraisal theory”. The change concerns wording only and does not affect the study’s objectives, data, or findings. We hope that the revised title improves the clarity and accuracy of the manuscript’s presentation.

We sincerely appreciate the constructive feedback from the editors and reviewers. The insightful comments have significantly improved the clarity, rigor, and theoretical contribution of our manuscript. We hope that the revisions have adequately addressed the concerns raised and that the manuscript is now suitable for publication in PLOS ONE.

Best regards,

Yuan Sun (on behalf of all co-authors)

Response to Reviewer 1:

Comment1:

The introduction does not sufficiently highlight the theoretical innovation of the study. While the introduction outlines the operational complexity of government accounts and mentions the dual attributes of policy advocacy and public service, it fails to clearly articulate how this research advances existing theoretical frameworks. For instance, it states that government accounts must "balance policy advocacy and public service responsibilities," but does not explicitly explain how integrating service quality into the SOR model—typically focused on information or system quality—creates a novel "dual-driver" framework for public sector communication. The concluding paragraph of the introduction should be revised to sharply contrast this study's integrated model (information + service quality) with prior SOR applications in commercial or UGC contexts, thereby underscoring its distinct theoretical contribution to understanding government social media effects.

We sincerely appreciate you for the insightful and constructive comments. Your suggestions provided valuable guidance for refining the theoretical positioning and rigor of the manuscript, and we have carefully revised the relevant sections accordingly.

Response1:

In response to your suggestions, we have systematically revised this section and further clarified the theoretical contributions and incremental value of our study in relation to existing research.

First, in the opening paragraph of the introduction, we elaborated on the composite functional characteristics of government culture and tourism TikTok accounts. We explicitly emphasize that such accounts not only serve as channels for tourism information dissemination and destination image construction, but also fulfill responsibilities related to public communication and online service responsiveness. This dual role reflects their institutional attributes and public responsibilities, distinguishing them from commercial or user-generated content (UGC) accounts.

Second, in the subsequent paragraph, we expanded the literature review by incorporating studies from both tourism short-video research and the e-government domain. We point out that prior studies have either focused on information content or platform-related technological attributes, or concentrated on administrative service-oriented e-government contexts. Relatively little attention has been paid to government social media accounts operated by culture and tourism departments, which simultaneously perform functions of “image promotion” and “public engagement.” Therefore, within this hybrid communication environment that integrates information dissemination and public engagement, the mechanisms through which service quality and information quality are associated with audience travel intentions remain insufficiently explored.

Third, at the end of the introduction, we added a concise statement highlighting the theoretical contributions of this study in three aspects. Initially, we extend the stimulus variables in the context of government social media tourism marketing by incorporating service quality as a relational factor into the analytical framework. Furthermore, we systematically explain how information quality and service quality are related to audience behavioral responses through distinct relational pathways, thereby constructing a more comprehensive dual-driving mechanism. Finally, at the organism level, we refine the processes of cognitive appraisal and emotional processing. This provides a clearer view of the internal psychological mechanisms. These processes explain how stimuli are associated with behavioral responses.

Comment2:

The literature review lacks critical synthesis and reads more like a summary list, failing to establish a clear dialogue with existing research. The review categorizes short-video marketing into three paradigms but does not critically analyze their limitations when applied to government-operated accounts. For example, it notes that most studies focus on information quality while overlooking service quality but does not engage with relevant public administration or e-government literature discussing the service role of official social media (e.g., Zhang et al., 2017 on government WeChat service quality). To strengthen the argument, the review should explicitly position itself against prior work, explaining why the service dimension is especially salient and under-theorized for government tourism accounts, and how this study's combined focus addresses a defined gap.

Response2:

Following your suggestion, we have substantially revised this section to strengthen the theoretical dialogue with existing studies.

First, in the second paragraph of the introduction, we restructured the review of tourism short video marketing studies. Through a systematic examination of prior research, we point out that most studies on social media tourism marketing primarily focus on stimulus factors such as information content characteristics, contextual cues, or platform-related technological attributes, while paying relatively limited attention to service quality as a relational dimension. At the same time, we analyzed relevant studies from the e-government field. Although e-government research emphasizes service-related elements such as reliability, responsiveness, and perceived trust, its core function is centered on administrative service delivery and efficiency improvement. This functional orientation differs fundamentally from the logic of government social media tourism marketing, which emphasizes image construction, emotional communication, and public engagement. Therefore, existing e-government service quality frameworks may not be directly applicable to the context of government culture and tourism social media marketing. Based on this analysis, government culture and tourism TikTok accounts serve both image promotion and public communication functions. In this dual context, the service dimension has both institutional and interactive significance, but remains insufficiently theorized. This gap constitutes a key entry point for our study.

Second, in the third paragraph of the introduction, we added an integrated discussion of the SOR model and the emotion appraisal theory. This addition provides a theoretical justification for incorporating both information quality and service quality into a unified analytical framework. On this basis, we propose a dual-path mechanism model to address the identified theoretical gap.

Furthermore, in the hypothesis development section, we refined and reorganized the relevant literature. By systematically reviewing prior findings and explicitly identifying their limitations, we proposed our research hypotheses in a more logically and critically synthesized manner.

Comment3:

The methods section is described too briefly, omitting key details necessary for replicability and assessment of validity. The adaptation process of measurement scales is only vaguely described as involving "translation and back-translation" and expert review. The manuscript should provide specific examples of how items were modified to fit the context of government Douyin accounts (e.g., original scale item vs. adapted item). Furthermore, the sampling procedure states that questionnaires were distributed "via WeChat and Douyin" but lacks details on the sampling frame, recruitment channels, or measures to prevent duplicate or inattentive responses (e.g., attention checks). Additionally, while Harman's single-factor test is mentioned, the result (38.93%) is not reported in the main text or tables. These omissions hinder the evaluation of the study's rigor.

Response3:

Following your suggestion, we have supplemented the Method section to provide clearer information on sampling procedures, measurement adaptation, and validity assessments, thereby enhancing the reproducibility and rigor of the research.

First, regarding data collection procedures, we have added detailed instructions on recruitment channels, screening criteria, and data quality control measures. To ensure that respondents had relevant experiential backgrounds, we provided a brief description of “government culture and tourism TikTok accounts” on the first page of the questionnaire and explicitly required that participants have prior experience viewing such short videos. The survey was administered through the Wenjuanxing platform, which was configured to prevent multiple submissions by blocking duplicate responses from the same login account, IP address, and device. Based on testing about response time, questionnaires completed in less than 90 seconds were considered invalid and removed from the dataset to minimize careless responses. In terms of recruitment channels, on WeChat the questionnaire was mainly distributed through tourism-interest groups and shared through appropriate snowball diffusion; on TikTok, we used private messages to invite participation, randomly selecting users who had commented on short videos posted by government culture and tourism accounts. These measures helped ensure better alignment between the sample and the research context.

Second, regarding common method bias, we implemented more rigorous statistical tests. In addition to reporting the Harman’s single-factor test result (with the first factor explaining 38.93% of the variance), we further introduced an unmeasured latent method factor into the baseline structural model to construct an extended model (Model M7). According to the criteria proposed by Chen (2007), changes in fit indices below 0.02 indicate no substantial improvement in model fit. Therefore, the inclusion of the unmeasured latent method factor did not significantly enhance model fit, suggesting that common method bias is unlikely to pose a serious threat to the findings. These results have now been clearly reported in the manuscript.

Third, we have added a comparison table in the Appendix comparing the original measurement scales with the adapted items.

Comment4:

4. The discussion of results is relatively superficial, primarily restating findings without deeply interpreting their theoretical significance or contextualizing them within the broader literature. For instance, the key finding that service quality exerts only an indirect effect (full mediation) while information quality has both direct and indirect effects is noted but not sufficiently explained. The discussion attributes this to "fragmented browsing" habits but does not connect it to theoretical mechanisms such as the heuristic processing of authoritative information versus the systematic evaluation required for service interactions. Moreover, the contrast with commercial tourism accounts—where service quality may have a direct impact—is hinted at but not substantiated with references to comparable studies. The discussion should explicitly link each major result back to the SOR and emotion appraisal frameworks, explore alternative explanations, and compare findings with related work to clarify the unique behavioral pathways in government social media contexts.

Response4:

Following your suggestion, we have substantially revised and expanded the discussion, with improvements made in the following aspects:

First, regarding the key finding that information quality exerts both direct and indirect effects, whereas service quality demonstrates only a fully mediated effect, we moved beyond the prior contextual explanation of “fragmented browsing habits.” Following your suggestion, we introduced the Heuristic–Systematic Model (HSM) as a complementary theoretical perspective to explain the underlying information-processing mechanisms. Specifically, information released by government culture and tourism accounts carries strong authority and official endorsement, which may trigger heuristic processing and thus directly predict travel intentions. In contrast, service quality involves perceptions of interaction, responsiveness, and relational evaluation, which require more systematic processing. Therefore, it is more likely to be indirectly associated with audience travel intentions through cognitive and emotional mediating mechanisms. This theoretical extension helps explain the structural differences in the action paths of the two types of stimulus variables.

Second, we strengthened dial

---

## [Decision Letter · Decision Letter 1]

15 Apr 2026

PONE-D-25-66483R1Understanding travel intention formation in government culture and tourism TikTok accounts: An integration of the SOR model and emotion appraisal theoryPLOS One

Dear Dr. Wen,

Thank you for submitting your manuscript to PLOS ONE. After careful consideration, we feel that it has merit but does not fully meet PLOS ONE’s publication criteria as it currently stands. Therefore, we invite you to submit a revised version of the manuscript that addresses the points raised during the review process.

We look forward to receiving your revised manuscript.

Kind regards,

Hai-Tao Yu, ph.D.

Academic Editor

PLOS One

Journal Requirements:

Reviewers' comments:

Reviewer's Responses to Questions

**Comments to the Author**

1. If the authors have adequately addressed your comments raised in a previous round of review and you feel that this manuscript is now acceptable for publication, you may indicate that here to bypass the “Comments to the Author” section, enter your conflict of interest statement in the “Confidential to Editor” section, and submit your "Accept" recommendation.

Reviewer #1: All comments have been addressed

Reviewer #2: (No Response)

2. Is the manuscript technically sound, and do the data support the conclusions?

Reviewer #1: Yes

Reviewer #2: Yes

3. Has the statistical analysis been performed appropriately and rigorously? 

Reviewer #1: Yes

Reviewer #2: Yes

4. Have the authors made all data underlying the findings in their manuscript fully available?

Reviewer #1: Yes

Reviewer #2: Yes

5. Is the manuscript presented in an intelligible fashion and written in standard English?

Reviewer #1: Yes

Reviewer #2: No

6. Review Comments to the Author

Reviewer #1: This manuscript addresses a timely and meaningful topic by examining how government culture and tourism TikTok accounts shape audience travel intentions. The study is well positioned at the intersection of tourism marketing, government social media communication, and short-video platform research, and it offers a clear attempt to extend the SOR framework through the integration of emotion appraisal theory. The paper is generally well organized, the theoretical framing is more focused than in many platform-based behavioral studies, and the empirical analysis is sufficiently solid to support the main claims. In particular, the distinction between information quality and service quality, as well as the identification of destination image perception and positive emotions as both parallel and sequential mediators, gives the paper a meaningful analytical structure. Overall, I find the manuscript publishable in its current form and recommend acceptance.

1. The topic selection is meaningful and has both theoretical and practical relevance. Rather than examining generic tourism short-video content or commercial accounts, the manuscript focuses specifically on government-operated culture and tourism TikTok accounts, which gives the study a clearer institutional context and a stronger public communication dimension. This is important because such accounts differ from ordinary commercial or user-generated content in that they combine destination promotion with public communication and online responsiveness. The paper therefore engages with a research object that is not only contemporary and policy-relevant, but also theoretically capable of generating new insights beyond mainstream tourism social media studies.

2. The language of the manuscript is generally fluent and appropriate for an international academic journal. The revised version reads much more professionally than a typical platform-behavior paper, and most sections are expressed in clear academic English with a relatively stable tone. Key constructs are defined in a readable way, the transitions between theory, hypotheses, methods, and results are mostly smooth, and the discussion section in particular shows improved precision in wording by avoiding overly strong causal claims in a cross-sectional design. This makes the manuscript easier to follow and improves its overall scholarly presentation.

3. The innovation and argumentation are clearly articulated. The paper does more than simply “apply SOR in a new setting.” Its main contribution lies in showing that information quality and service quality operate through different relational pathways in the context of government tourism social media. The finding that information quality has both direct and indirect associations with travel intention, whereas service quality operates through full mediation, is theoretically interesting and is explained in a reasonably convincing way through the added heuristic–systematic processing perspective. In addition, the manuscript refines the organism component by distinguishing cognitive appraisal from emotional response and by empirically supporting a cognition-to-emotion progression. This gives the theoretical model a stronger internal logic and helps the paper stand out from many studies that treat mediators in a more generic or interchangeable manner.

4. The article provides sufficiently developed discussion and does not stop at merely reporting statistical significance. The revised discussion links the main findings back to the SOR framework and emotion appraisal theory, and it also attempts to explain why service quality functions differently in a government communication context than it might in a commercial setting. I also appreciate that the authors compare their findings with prior studies, discuss the ordering between destination image perception and positive emotions, and derive managerial implications that are aligned with the empirical results rather than being overly generic. The move from cognitive construction to emotional reinforcement is especially well connected to the reported mediation structure.

5. The manuscript is methodologically and formally规范 in ways that support confidence in the findings. The authors provide details on questionnaire adaptation, sampling procedures, recruitment channels, duplicate-response controls, and response-time screening. They report reliability and validity indicators, compare competing CFA models, add SRMR, and supplement Harman’s single-factor test with an unmeasured latent method factor approach. The reporting of sample characteristics, measurement items, ethics statement, conflict-of-interest statement, funding information, and data availability is also relatively complete. These features give the paper a level of formal rigor and transparency that is appropriate for publication.

Reviewer #2: The manuscript investigates the formation of audience travel intentions in the context of government culture and tourism TikTok accounts (GCTTAs) by integrating the Stimulus–Organism–Response (SOR) model with emotion appraisal theory. The topic is timely and relevant, particularly given the increasing role of government social media in destination marketing and public communication.

Overall, the manuscript is well-structured, and the research design is generally sound. The integration of information quality and service quality within a unified analytical framework, along with the examination of destination image perception and positive emotions as mediators, provides a meaningful contribution to understanding behavioral mechanisms in this context. The empirical analysis is clearly presented, and the results are coherent and interpretable.

However, several issues should be addressed to further strengthen the manuscript:

First, the theoretical contribution could be further sharpened. While the manuscript highlights its contributions more clearly than before, part of the contribution still appears to be framed as applying existing theories to a new context. The authors are encouraged to more explicitly articulate how the findings—particularly the fully mediated role of service quality—extend or refine existing theoretical perspectives, especially in relation to service quality and public-sector communication.

Second, although the manuscript distinguishes between information quality and service quality conceptually, some measurement items related to service quality may overlap with content or platform characteristics. Additional clarification would help strengthen the conceptual distinctiveness between these constructs and improve construct validity.

Third, the discussion of the non-significant direct effect of service quality on travel intentions could be further deepened. This finding is potentially important, and the authors may consider elaborating on alternative explanations, such as the role of trust-building, differences between immediate and long-term effects, or characteristics of short-video consumption environments.

Fourth, while the manuscript includes robustness checks such as competing model comparisons and common method bias tests, additional robustness analyses (e.g., alternative model specifications or subgroup analysis) would further enhance confidence in the findings.

Fifth, regarding data availability, the manuscript states that all relevant data are provided within the manuscript and its Supporting Information files, which is consistent with the journal’s policy. The authors are encouraged to ensure that the dataset is clearly documented (e.g., variable definitions and coding schemes) to facilitate reproducibility.

Finally, the manuscript is generally understandable, but minor grammatical errors and awkward phrasing remain. A final round of professional English editing is recommended to improve clarity and readability.

In summary, the manuscript has good potential for publication. With the above revisions, it would be suitable for acceptance.

7. PLOS authors have the option to publish the peer review history of their article (what does this mean?). If published, this will include your full peer review and any attached files.

Reviewer #1: No

Reviewer #2: **Yes:** Yuan Li

---

## [Author Response · Author response to Decision Letter 2]

27 Apr 2026

Response Letter

Dear Editor and Reviewers,

We would like to sincerely thank you for your constructive and insightful comments on our manuscript entitled "Understanding travel intention formation in government culture and tourism TikTok accounts: An integration of the SOR model and emotion appraisal theory" (ID:PONE-D-25-66483R1). We greatly appreciate the time and effort you have devoted to reviewing our work. Your suggestions have been invaluable in helping us improve the clarity, rigor, and theoretical contribution of the manuscript.

We have carefully revised the manuscript to ensure that all changes are clearly presented, and we hope that the revisions adequately address your comments. The main revisions are summarized as follows:

1.Redundant expressions have been removed throughout the manuscript to improve conciseness and clarity.

2.Typographical errors, grammatical issues, and awkward phrasing have been revised to enhance readability and academic rigor.

3.Terminology has been standardized across the manuscript. For example, terms such as “government departments” and “public sector” have been consistently revised to “government culture and tourism departments” to ensure conceptual consistency.

4.The definition of service quality has been revised to establish a clearer conceptual distinction from information quality.

5.The theoretical contributions have been further refined and strengthened. In particular, the full mediation effect of service quality has been elaborated, highlighting its context-dependent nature, whereby its role is jointly shaped by the characteristics of information sources, communication objectives, and social media platform features.

6.Additional robustness checks have been conducted using subgroup analysis to further enhance the reliability of the findings.

7.The discussion of the non-significant direct association between service quality and travel intention has been further developed. Following the reviewers’ suggestions, alternative explanations have been incorporated, including the role of trust development, differences between short-term and long-term effects, and the characteristics of short-video consumption contexts.

We believe that these revisions have improved the overall quality and rigor of the manuscript. We sincerely appreciate the time and constructive feedback provided by the editor and reviewers, and we look forward to your further evaluation.

Sincerely,

Yuan Sun (on behalf of all co-authors)

Response to Reviewer 1:

Comment1:

Reviewer #1: This manuscript addresses a timely and meaningful topic by examining how government culture and tourism TikTok accounts shape audience travel intentions. The study is well positioned at the intersection of tourism marketing, government social media communication, and short-video platform research, and it offers a clear attempt to extend the SOR framework through the integration of emotion appraisal theory. The paper is generally well organized, the theoretical framing is more focused than in many platform-based behavioral studies, and the empirical analysis is sufficiently solid to support the main claims. In particular, the distinction between information quality and service quality, as well as the identification of destination image perception and positive emotions as both parallel and sequential mediators, gives the paper a meaningful analytical structure. Overall, I find the manuscript publishable in its current form and recommend acceptance.

1. The topic selection is meaningful and has both theoretical and practical relevance. Rather than examining generic tourism short-video content or commercial accounts, the manuscript focuses specifically on government-operated culture and tourism TikTok accounts, which gives the study a clearer institutional context and a stronger public communication dimension. This is important because such accounts differ from ordinary commercial or user-generated content in that they combine destination promotion with public communication and online responsiveness. The paper therefore engages with a research object that is not only contemporary and policy-relevant, but also theoretically capable of generating new insights beyond mainstream tourism social media studies.

2. The language of the manuscript is generally fluent and appropriate for an international academic journal. The revised version reads much more professionally than a typical platform-behavior paper, and most sections are expressed in clear academic English with a relatively stable tone. Key constructs are defined in a readable way, the transitions between theory, hypotheses, methods, and results are mostly smooth, and the discussion section in particular shows improved precision in wording by avoiding overly strong causal claims in a cross-sectional design. This makes the manuscript easier to follow and improves its overall scholarly presentation.

3. The innovation and argumentation are clearly articulated. The paper does more than simply “apply SOR in a new setting.” Its main contribution lies in showing that information quality and service quality operate through different relational pathways in the context of government tourism social media. The finding that information quality has both direct and indirect associations with travel intention, whereas service quality operates through full mediation, is theoretically interesting and is explained in a reasonably convincing way through the added heuristic–systematic processing perspective. In addition, the manuscript refines the organism component by distinguishing cognitive appraisal from emotional response and by empirically supporting a cognition-to-emotion progression. This gives the theoretical model a stronger internal logic and helps the paper stand out from many studies that treat mediators in a more generic or interchangeable manner.

4. The article provides sufficiently developed discussion and does not stop at merely reporting statistical significance. The revised discussion links the main findings back to the SOR framework and emotion appraisal theory, and it also attempts to explain why service quality functions differently in a government communication context than it might in a commercial setting. I also appreciate that the authors compare their findings with prior studies, discuss the ordering between destination image perception and positive emotions, and derive managerial implications that are aligned with the empirical results rather than being overly generic. The move from cognitive construction to emotional reinforcement is especially well connected to the reported mediation structure.

5. The manuscript is methodologically and formally in ways that support confidence in the findings. The authors provide details on questionnaire adaptation, sampling procedures, recruitment channels, duplicate-response controls, and response-time screening. They report reliability and validity indicators, compare competing CFA models, add SRMR, and supplement Harman’s single-factor test with an unmeasured latent method factor approach. The reporting of sample characteristics, measurement items, ethics statement, conflict-of-interest statement, funding information, and data availability is also relatively complete. These features give the paper a level of formal rigor and transparency that is appropriate for publication.

Response:

We sincerely thank you for the positive and encouraging evaluation of our manuscript. We greatly appreciate your recognition of the study’s theoretical contributions, methodological rigor, and clarity of presentation. At the same time, we have carefully proofread the entire manuscript to further improve its overall quality and clarity of expression.

Thank you again for your valuable and constructive comments, the time you have devoted, and your support for our study.

Response to Reviewer 2:

Comment1:

The manuscript investigates the formation of audience travel intentions in the context of government culture and tourism TikTok accounts (GCTTAs) by integrating the Stimulus–Organism–Response (SOR) model with emotion appraisal theory. The topic is timely and relevant, particularly given the increasing role of government social media in destination marketing and public communication.

Overall, the manuscript is well-structured, and the research design is generally sound. The integration of information quality and service quality within a unified analytical framework, along with the examination of destination image perception and positive emotions as mediators, provides a meaningful contribution to understanding behavioral mechanisms in this context. The empirical analysis is clearly presented, and the results are coherent and interpretable.

However, several issues should be addressed to further strengthen the manuscript:

First, the theoretical contribution could be further sharpened. While the manuscript highlights its contributions more clearly than before, part of the contribution still appears to be framed as applying existing theories to a new context. The authors are encouraged to more explicitly articulate how the findings—particularly the fully mediated role of service quality—extend or refine existing theoretical perspectives, especially in relation to service quality and public-sector communication.

Response1:

We sincerely appreciate you for the insightful and constructive comments. Your suggestions provided valuable guidance for refining the theoretical positioning and rigor of the manuscript, and we have carefully revised the relevant contents accordingly.

We agree that merely applying existing theories to a new context is insufficient to establish a strong theoretical contribution. Following your suggestion, we have further strengthened the theoretical contribution of this study.

Specifically, our findings indicate that, in the context of social media marketing by government culture and tourism departments, service quality primarily operates through cognition–emotion pathways, rather than exerting a direct influence on audiences’ travel intentions. This fully mediated effect challenges the commonly held assumption in prior research that service quality directly affects behavioral outcomes. Furthermore, we emphasize that the role of service quality is contingent upon multiple factors, including the nature of the information source, communication objectives, and the characteristics of short-video platforms. This study provides a more nuanced understanding of the mechanisms through which service quality operates in public-sector tourism communication.

Comment2:

Second, although the manuscript distinguishes between information quality and service quality conceptually, some measurement items related to service quality may overlap with content or platform characteristics. Additional clarification would help strengthen the conceptual distinctiveness between these constructs and improve construct validity.

Response2:

Thank you for this insightful and important comment. We agree that clearly distinguishing service quality from information quality is essential for improving construct validity.

Following your suggestion, we have further strengthened the conceptual distinction between these two constructs in the manuscript. Specifically, we clarify that information quality refers to the evaluation of content attributes of short videos, whereas service quality is conceptualized as an account-level construct that captures audiences’ perceptions of the overall performance of government tourism departments in delivering information and services via the TikTok platform.

In addition, we have refined the definitions of the three dimensions of service quality (reliability, responsiveness, and assurance), emphasizing their interaction-oriented and service-oriented nature rather than content-related attributes, thereby avoiding potential conceptual overlap with information quality.

We hope that the above revisions help establish a clearer conceptual boundary between the two constructs and further enhance construct validity.

Comment3:

Third, the discussion of the non-significant direct effect of service quality on travel intentions could be further deepened. This finding is potentially important, and the authors may consider elaborating on alternative explanations, such as the role of trust-building, differences between immediate and long-term effects, or characteristics of short-video consumption environments.

Response3:

Following your suggestion, we have further expanded the discussion of this finding from multiple perspectives.

First, from the evaluative nature of service quality, we note that such evaluations rely on continuous interaction and feedback, making them difficult to form through cognitive shortcuts in a single exposure. Based on the Heuristic–Systematic Model (HSM), service quality is therefore more likely to operate through the systematic processing route rather than serving as an immediate heuristic cue to influence behavioral intentions.

Second, we introduce a temporal perspective by distinguishing between short-term and long-term effects, emphasizing that the influence of service quality unfolds through an accumulation process.

Third, considering the characteristics of short-video consumption environments, we point out that algorithm-driven recommendation mechanisms and fast-paced content exposure reinforce users’ reliance on heuristic processing. This, in turn, constrains the immediate impact of service quality, which depends on systematic evaluation.

Comment4:

Fourth, while the manuscript includes robustness checks such as competing model comparisons and common method bias tests, additional robustness analyses (e.g., alternative model specifications or subgroup analysis) would further enhance confidence in the findings.

Response4:

Following your suggestion, we have conducted additional robustness analyses by implementing multi-group structural equation modeling (MGSEM) to examine the stability of the proposed model across different subgroups. Specifically, we performed multi-group invariance tests based on gender, age, and usage frequency.

Following established procedures, we assessed configural, metric, and structural invariance by comparing a sequence of nested models with increasingly restrictive parameter constraints. The results (Table 4) indicate that the unconstrained baseline models demonstrate acceptable fit across all groups. As constraints were progressively imposed, the changes in model fit indices (e.g., ∆CFI) remained minimal and well within recommended thresholds.

Although the chi-square difference test is statistically significant in a few cases, the corresponding ∆CFI values remain below the critical value of 0.01, suggesting that the deterioration in model fit is negligible and that the invariance assumption is not violated.

Overall, the findings from the multi-group analyses demonstrate that the measurement and structural relationships are stable across gender, age, and usage frequency groups. This provides additional evidence for the robustness of the proposed model and supports the generalizability of the results across different demographic and behavioral segments.

Comment5:

Fifth, regarding data availability, the manuscript states that all relevant data are provided within the manuscript and its Supporting Information files, which is consistent with the journal’s policy. The authors are encouraged to ensure that the dataset is clearly documented (e.g., variable definitions and coding schemes) to facilitate reproducibility.

Response5:

The questionnaire instrument and the complete dataset have already been provided in the manuscript and the Supporting Information files. The dataset is based directly on the original measurement scales, and all variables correspond closely to the questionnaire items. Therefore, the data structure is transparent and straightforward, enabling straightforward interpretation and reproducibility without the need for additional coding schemes.

In addition, we have carefully reviewed the dataset and the corresponding descriptions in the manuscript to ensure consistency and clarity, so that the analysis can be easily reproduced.

We hope that the current level of documentation is sufficient to support tr

---

## [Editor Report · Decision Letter 2]

28 Apr 2026

Understanding travel intention formation in government culture and tourism TikTok accounts: An integration of the SOR model and emotion appraisal theory

PONE-D-25-66483R2

Dear Dr. Wen,

We’re pleased to inform you that your manuscript has been judged scientifically suitable for publication and will be formally accepted for publication once it meets all outstanding technical requirements.

Kind regards,

Hai-Tao Yu, ph.D.

Academic Editor

PLOS One
---

## [Editor Report · Acceptance letter]

PONE-D-25-66483R2

PLOS One

Dear Dr. Wen,

I'm pleased to inform you that your manuscript has been deemed suitable for publication in PLOS One. Congratulations! Your manuscript is now being handed over to our production team.

Kind regards,

on behalf of

Dr. Hai-Tao Yu

Academic Editor

PLOS One